# The Short-Term Price Elasticity, Temperature Elasticity, and Wind Speed Elasticity of Electricity: A Case Study from Norway

Johannes Idsø *,†, Jon Gunnar Nesse † and Øyvind Heimset Larsen †

Western Norway Research Institute, P.O. Box 163, 6851 Sogndal, Norway; post@vestforsk.no (J.G.N.); ohl@vestforsk.no (Ø.H.L.)
* Correspondence: johannes.idso@hvl.no; Tel.: +47-400-60-153
† These authors contributed equally to this work.

**Abstract:** Energy production using hydropower has a 150-year history in Norway. High mountains, lots of rain, and a well-developed technology laid the foundation for low and stable electricity prices. The Norwegian electricity market is unique and different from any other country. Nearly all electricity produced (98.3 percent) comes from renewable energy sources and 75 percent of the energy used for heating is electricity. From autumn 2020, major changes have been observed in the electricity market in Norway. In 2021, Norway opened two transmission cables, one to Germany and one to England. Both cables have a capacity of 1400 MW. The average price per MWh was NOK 263 in southern Norway in the period 2013–2020, which more than quadrupled to NOK 1192 per MWh in the period 2021–2023. We have investigated how the market reacted to the large price increase. We found that price elasticity is low even when the price is very high. It is the temperature that controls the consumption. When it is cold—below zero degrees Celcius—the temperature elasticity is close to zero; the temperature elasticity is not constant. When the temperature is above zero, the temperature elasticity is about −0.7. Price variations or changes in wind speed only lead to minor adjustments in electricity consumption. It is the variations in temperature that result in the observable fluctuations in electricity consumption. Since Norway exports electricity to Sweden, Denmark, Finland, Germany, the Netherlands, and England, knowledge of the Norwegian electricity market is relevant for many market participants. The Norwegian electricity market differs from those in other countries. Therefore, there is a risk that conclusions drawn about the Norwegian electricity market based on research conducted in other countries may be incorrect or inaccurate. Our contribution with this case study is to deepen the knowledge of how the electricity market in Norway operates.

**Keywords:** price elasticity of electricity; temperature elasticity of electricity; wind speed elasticity of electricity; consumption of electricity; transmission cables of electricity

## 1. Introduction

Concerns related to climate change as a consequence of increased $CO_2$ levels in the atmosphere have led many countries to adopt goals to reduce emissions of $CO_2$ and other greenhouse gases. The practical implementation of these emission targets has resulted in structural changes in industries and shifts in the technologies used for electricity production and other industries [1]. This, in turn, has led to changes in the energy market in Northern Europe. For example, Germany and the United Kingdom have expanded their capacity for renewable electricity production, particularly seeing significant increases in wind and solar power production. However, the variability of wind and solar power poses a challenge, necessitating a backup system. Given that Norway can produce more electricity than it needs in most years, it can contribute to stabilizing the supply side of the electricity market in Northern Europe with its stable hydropower [2].

However, Norway also has its climate targets. It has been decided that oil production, which currently accounts for a significant percentage of the country's $CO_2$ emissions, will

cease using generators powered by fossil fuels to produce electricity for oil platforms. Instead, the required electricity will come from offshore wind farms or cables from the mainland. Additionally, Norway aims to develop industries within battery production, hydrogen production, and ammonia production based on renewable energy [3].

International needs and national goals have led to changes in the Norwegian energy market. To make optimal decisions, it is necessary to understand how the Norwegian electricity market functions. This article contributes to that understanding.

How does the electricity market operate? It is challenging to provide a universal answer to this question, as the market varies significantly across different regions, influenced by cultural, business, and climatic factors unique to each country. For this reason, many articles and papers have been published that deal with the production or consumption of electricity in a number of countries [4–7].

However, certain events within a country can draw widespread interest. For example, in the autumn of 2020, electricity prices in southern Norway escalated by more than 400 percent compared to the average of the preceding seven years. This high price trend continued through 2021 and 2022 and into the first half of 2023. The substantial and sustained increase in electricity prices in southern Norway during autumn 2020 presents an opportunity to investigate the market's response to unprecedented price hikes.

The following sections will cover the development of electricity production in Norway, how the price of electricity is set in the Nordic countries, and how consumption in Norway varies throughout the day, week, and year. We also describe how trade in electricity between Norway and other countries has evolved from the first cables in the 1960s to 2023. We show how consumption in Norway developed from 2013 to 2023. After Norway received the last two cables in 2021, the price of electricity in southern Norway has significantly increased and has been highly variable.

The Nordic countries have cold winters, and the need for energy to heat homes is great [8]. In our subsequent analysis, we will develop a model where electricity demand is determined by price, temperature, and wind speed, and we will assess the associated elasticities. Furthermore, we will explore whether price and temperature elasticities remain stable or vary in response to changes in price and temperature.

### 1.1. Electricity Production in Norway

Norway's high mountains and location along the rainy west coast of Eurasia make it an ideal setting for hydropower production [9]. The first hydropower plants were constructed in the 1880s [10,11], and as of February 2023, there were 1761 operational power plants. As illustrated in Figure 1, there has been a significant increase in power production since 1950. Since production is weather-dependent, annual output fluctuates. In 2021, Norway achieved its highest ever production at 157 TWh. The average production from 2018 to 2022 was 148 TWh [12].

There are two primary benefits of using hydropower for electricity generation. Firstly, production can be easily regulated, which is crucial since production and consumption need to be in constant balance. This flexibility is particularly beneficial in supplementing neighboring countries like Germany and England, especially during periods when their wind power generation is low due to lack of wind [13,14].

The second advantage is the minimal wear and tear on the production equipment, resulting in low production costs. The estimated average cost for all hydropower plants in Norway in 2023 was NOK 117.7 per MWh, which is approximately EUR 10 per MWh [15].

Although the construction of hydroelectric power plants in Norway began before 1900, it was not until after 1950 that there was a significant expansion in hydropower development.

Despite challenges associated with variable production and the fluctuating voltage of electricity from this source [16], wind power plants have been constructed since 1998. Nevertheless, hydropower remains the predominant technology [17]. It is important that the technology used for energy production has local acceptance [18].

In 2023, 89.2 percent of Norway's electricity production was generated from hydropower, with the remainder coming from wind power (9.1 percent) and thermal production (1.7 percent) [12]. This means that 98.3 per cent of the electricity came from renewable energy sources.

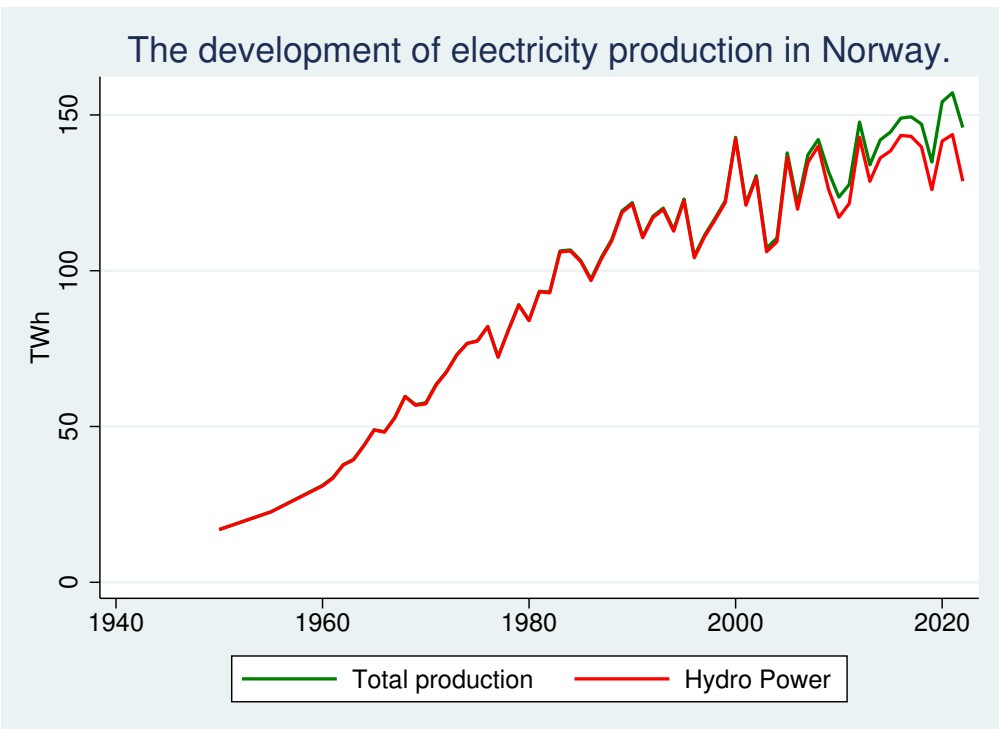

**Figure 1.** The development of electricity production in Norway since 1950. Since almost all production is based on water or wind, production will vary from year to year depending on rainfall and wind. Until the year 2000, the total production was equal to the hydropower production.

The majority of large hydropower plants in Norway are publicly owned. Municipalities, county municipalities, and the state collectively hold ownership of approximately 90 percent of the hydropower production capacity in the country [19]. Private landowners are typically granted licenses to build smaller power plants. In 2023, the largest privately owned plant has a capacity of 23.7 MW [20].

There are two main types of hydropower plants in Norway: river power plants, which operate without reservoirs, and those equipped with reservoirs. River power plants have limited flexibility in regulating production due to their design.

Figure 2 illustrates the variations in water levels, and consequently, the changes in stored energy. During winter, most precipitation falls as snow, leading to minimal water inflow into the reservoirs. The energy production in winter and early spring depletes the reservoirs. As spring progresses, snowmelt increases, and the inflow from melting snow eventually exceeds the water used for production. This causes reservoir levels to rise throughout spring and autumn. The crucial task for power producers is to manage water resources judiciously, ensuring that there is sufficient water in the reservoirs to sustain production through the winter.

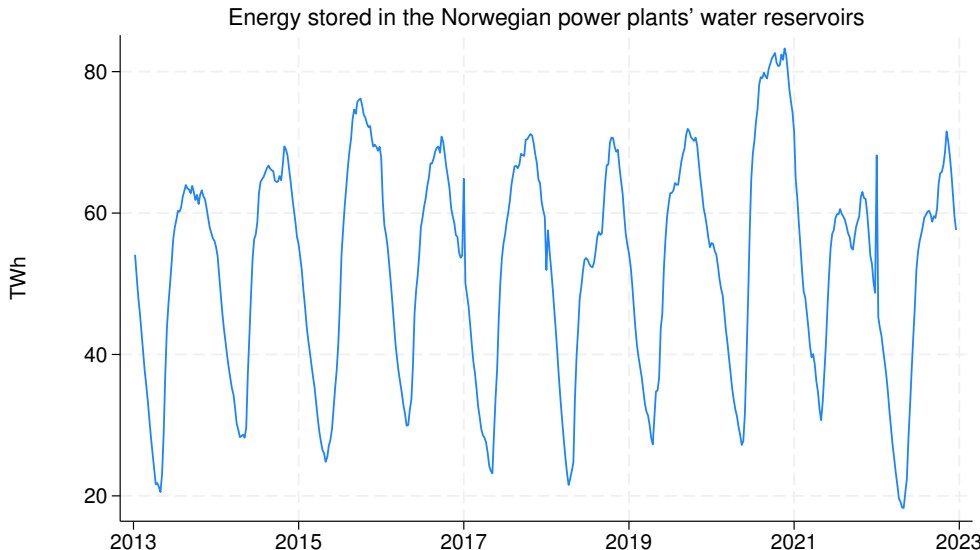

**Figure 2.** Energy stored in the Norwegian power plants' water reservoirs measured in TWh. The water reservoirs of power plants are at their lowest in the spring before snowmelt leads to the inflow of water being higher than what is used for production. The highest level is in the fall before precipitation comes as snow.

### 1.2. The Pricing of Electricity in Norway and the Nordic Countries

Prior to the implementation of the Energy Act in 1990, electricity in Norway was sold at the cost of production, resulting in low prices set by the authorities. This led to businesses and households becoming accustomed to low and stable electricity costs.

The Energy Act of 1990 marked a significant shift, introducing a market-based system for the sale and purchase of electricity in Norway [21]. This law established a competitive market, where electricity prices were determined by supply and demand, which significantly modernized Norway's energy sector. It also laid the groundwork for the creation of Nord Pool.

Nord Pool, initially formed in 1996 through cooperation among Nordic countries, involved the national grid companies of Norway (Statnett), Sweden (Svenska Kraftnät), Finland (Fingrid), and Denmark (Energinet.dk). This collaboration aimed to develop a more integrated and efficient electricity market across the Nordics.

As a result of the Energy Act, nearly all electricity sold in Norway is traded through Nord Pool, Europe's leading power market. Nord Pool facilitates trading, clearing, settlement, and related services in both day-ahead and intraday markets across 16 European countries [22].

Nord Pool establishes the market price as follows: Producers submit to Nord Pool their planned production volumes at varying prices for the next 24 h. For producers, particularly those using water from reservoirs, the opportunity cost—potential future income from the water if conserved for later production—influences their supply function more than the immediate production costs.

Similarly, Nord Pool gathers information on expected demand. Using these data on supply and demand, Nord Pool calculates the equilibrium electricity price for different price areas in the connected countries. The price for the next 24 h is announced at 1 p.m. the day before.

### 1.3. The Consumption of Electricity

The consumption of electricity mirrors the rhythm of society, fluctuating in sync with business activities. Each enterprise, through its operations, contributes to the aggregate demand for electricity. Likewise, individual actions, such as using lights and electrical appliances, while seemingly minor, collectively play a crucial role in this dynamic landscape.

Cultural factors significantly influence electricity consumption, both in private and business spheres. Different societies, with their unique customs and traditions, shape when and how electricity is used.

Moreover, climatic conditions are pivotal in determining overall electricity usage. Regions with cold winters often see a surge in electricity demand for heating, while hot summers can lead to increased air conditioning use. Thus, electricity demand presents a complex interplay of economic activities, personal habits, cultural norms, and environmental factors.

Electricity consumption exhibits three primary patterns of fluctuation: daily, weekly, and yearly. Figure 3 illustrates the daily consumption patterns in relation to the average. There is a notable variation in these patterns between summer and winter. During winter, the lowest electricity usage typically occurs between 2–3 a.m., peaking between 4–5 p.m. In contrast, summer consumption reaches its lowest between 3–4 a.m., with the highest demand occurring between 10–11 a.m.

There is also a significant seasonal variation in daily electricity use. For instance, electricity consumption in July is only about 55 percent of that in January, highlighting the substantial differences between summer and winter demand.

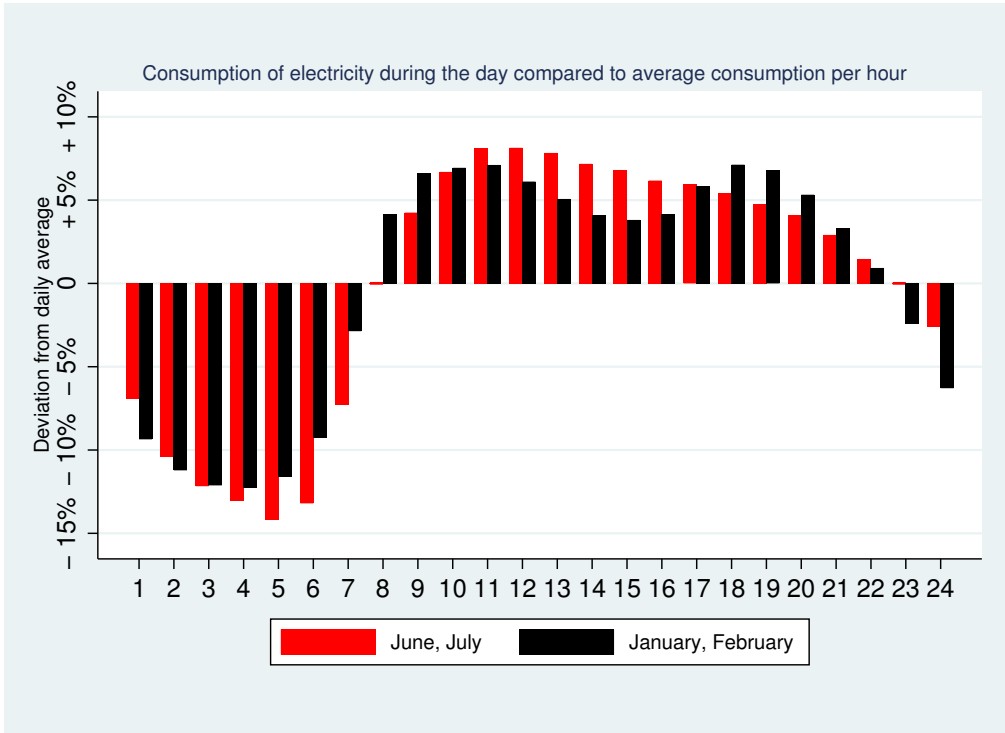

**Figure 3.** Consumption over the 24 h of the day compared to the average consumption over the day in the years 2013–2022.

Figure 4 illustrates the weekly pattern of electricity consumption. In this depiction, the average consumption on each day is compared to the overall average daily consumption for the week. Notably, Saturdays and Sundays experience the lowest levels of electricity use. However, the disparity between normal weekdays and the weekend (Saturday/Sunday) is more pronounced in summer than in winter, highlighting a clear variation in consumption patterns between these seasons.

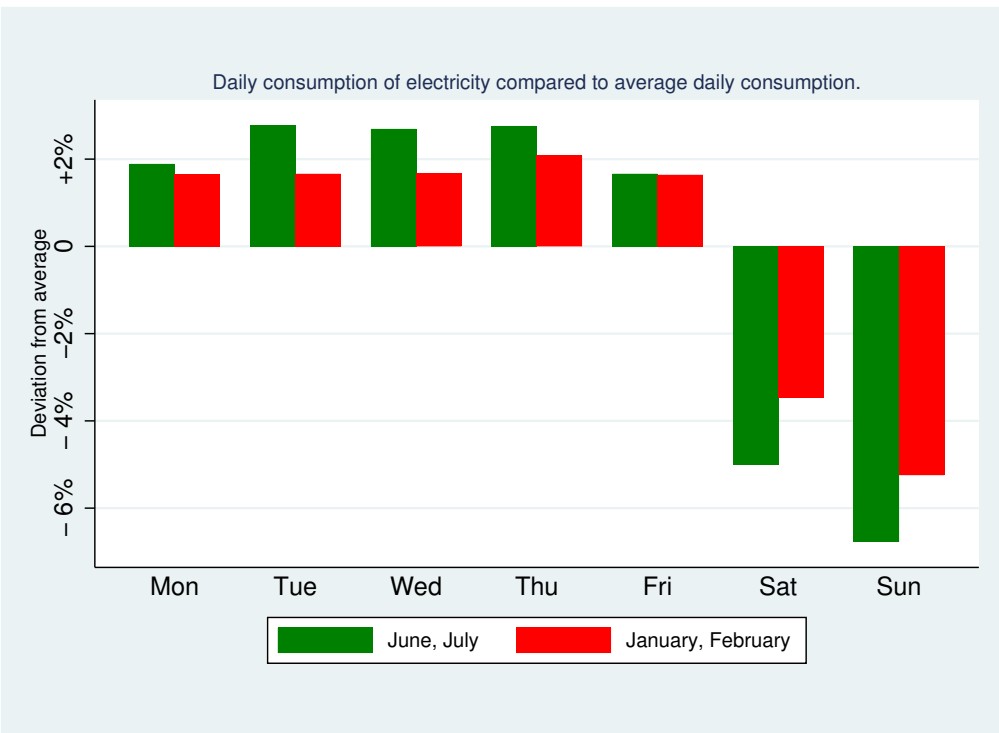

**Figure 4.** Consumption on the individual days of the week compared to the average consumption per day of the week.

Figure 5 shows the consumption pattern over the year. In the figure, the average consumption in the individual months is compared with the average monthly consumption for the whole year.

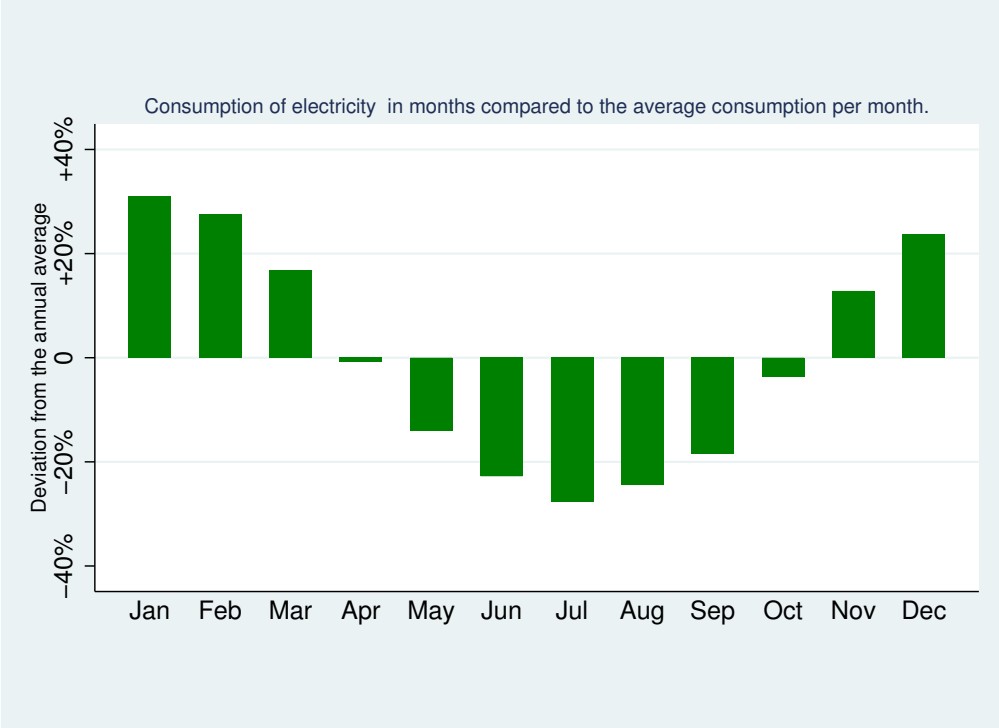

**Figure 5.** Consumption in the individual months compared to the average consumption per month during the year.

In Norway, over 80 percent of the energy consumed in homes comes from electricity, which is a stark contrast to the EU average of around 25 percent [12]. The primary reason for this high consumption in Norway is its climate, characterized by cold winters with temperatures frequently dropping below $-20\,°C$. Electricity in Norway is extensively used for heating residential homes, commercial properties, and public buildings.

### 1.4. Consumption over Several Years

Cultural, climatic, and economic conditions influence the consumption of electricity from hour to hour with remarkable pattern stability over time. Figure 6 shows the daily consumption over the year from and including January 2013 to January 2023. From the figure, we see that consumption over the year fluctuates as a stationary time series.

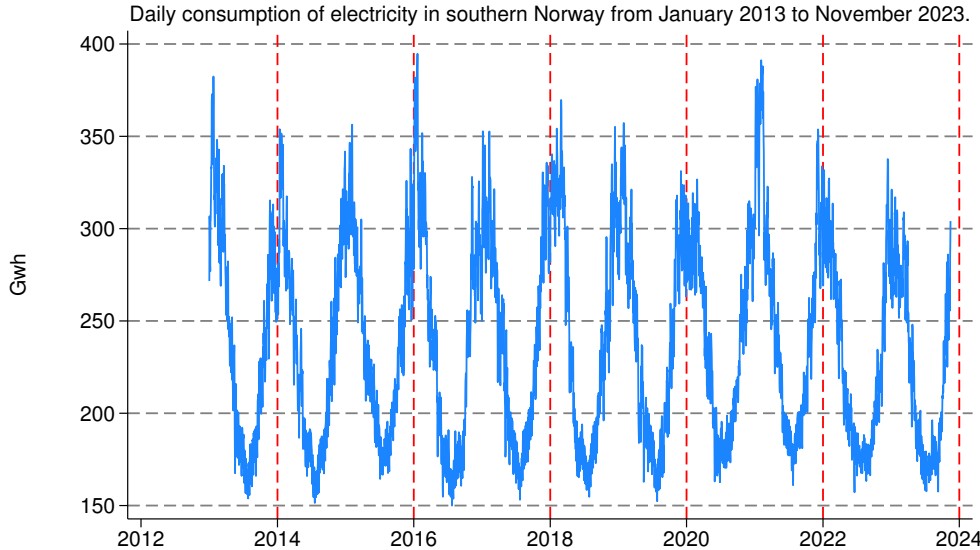

**Figure 6.** Average consumption of electricity per day from 2013 to November 2023 in southern Norway.

Figure 6 indicates that the consumption of electricity can be divided into a fixed and a variable part. In this case, where we are studying consumption in southern Norway, we see that the daily consumption is never below 150 GWh, while daily consumption above 150 GWh fluctuates over the year. It is these fluctuations that we study.

### 1.5. Trade in Electricity

Norway has been engaged in electricity trade with other countries since 1960, starting with its first cable connection to Sweden. Over time, Norway has increasingly integrated into the Northern European electricity market. As of November 2023, Norway has established eight cables to Sweden, four to Denmark, and one each to Finland, the Netherlands, Russia, Germany, and England [23]. The cable connecting to Germany became operational in March 2021, followed by the one to England in October of the same year. Both cables have a capacity of 1400 MW each. Figure 7 illustrates the electricity trading dynamics.

Usually, the production of electricity in Norway is greater than the consumption. As Figure 8 shows, in 23 of the 32 years from the year 1990 to the year 2022 the hydropower production in Norway was larger than the consumption of electricity. In the period 2000–2022, the total net export of electricity from Norway to other northern European countries was 191 TWh [24].

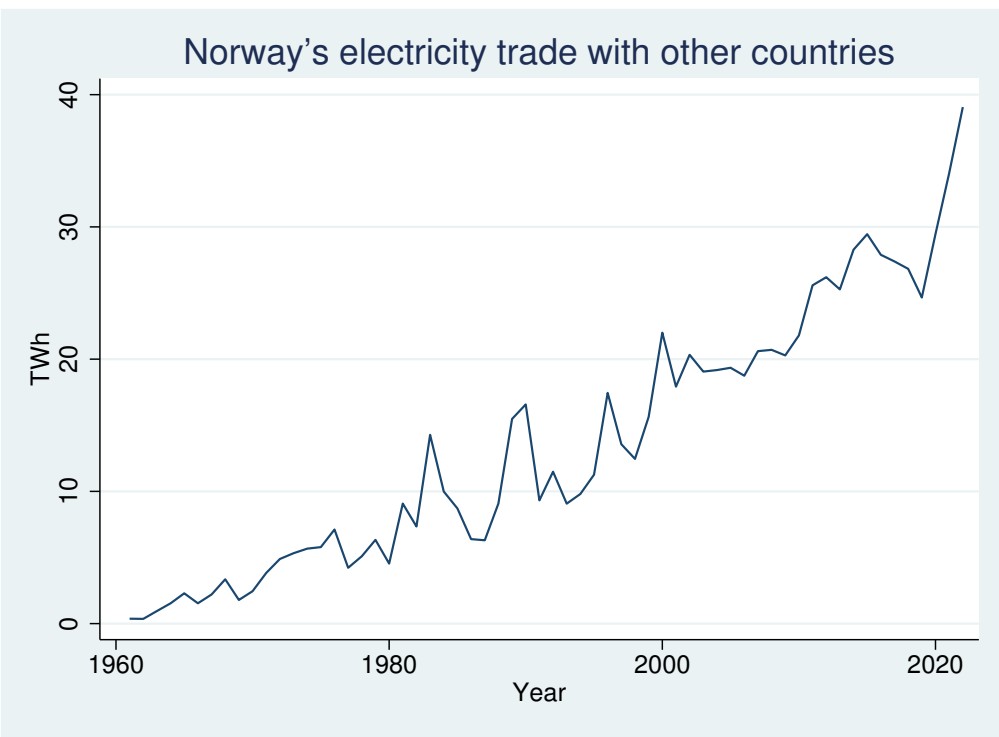

**Figure 7.** Trade in electricity. The sum of import and export since 1960.

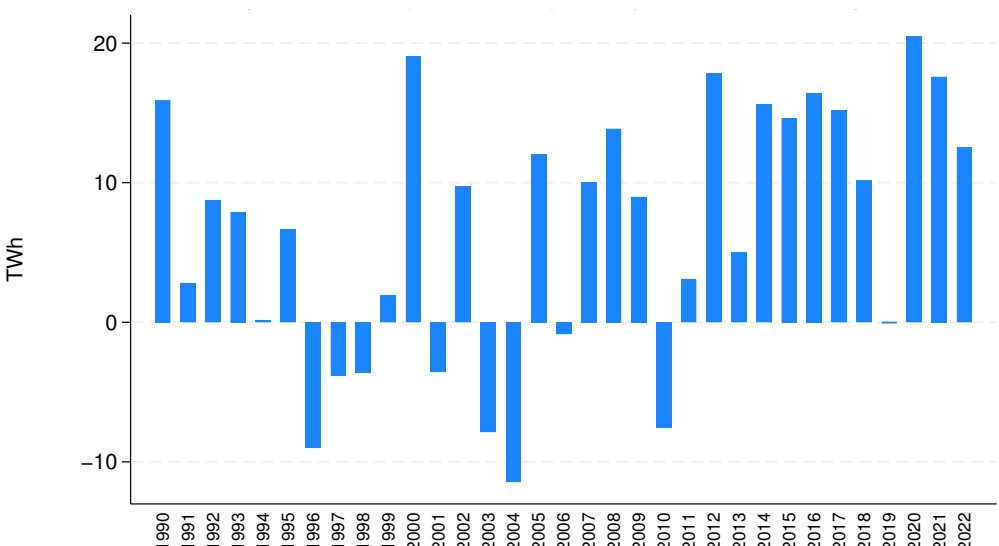

**Figure 8.** The net export of electricity from Norway in the years 1990–2022. Figures in Twh.

As can be seen from Figure 9, since 1960, the exchange of electricity with other countries has had a development that largely coincides with production.

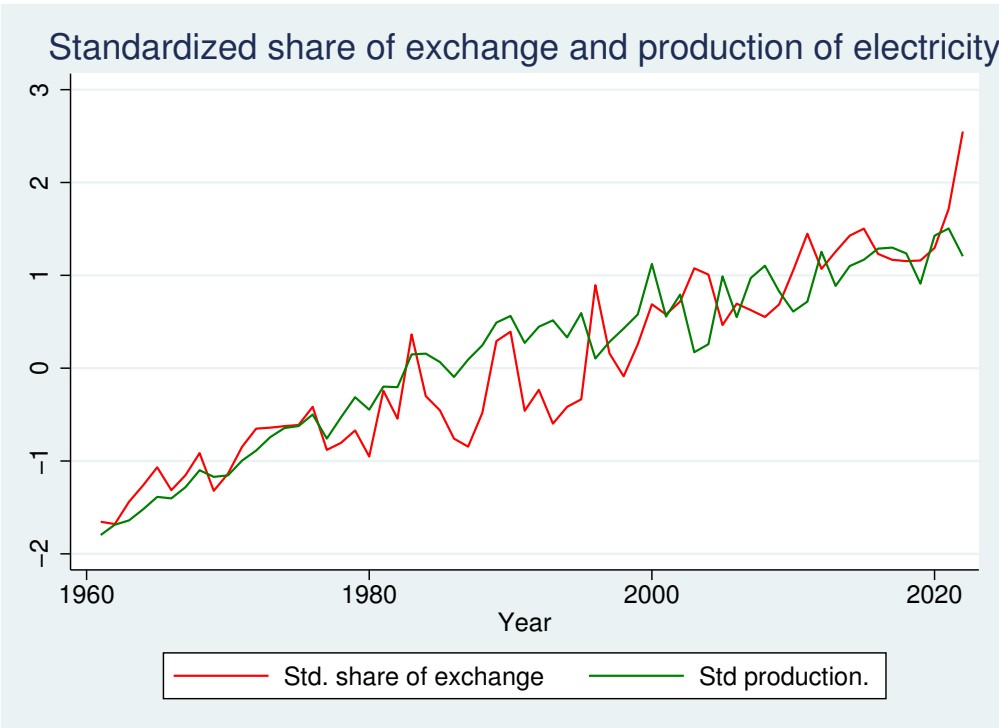

**Figure 9.** Standardized share of exchange and Standardized production of electricity.

### 1.6. Electricity Prices in Norway

The special feature of the electricity market is that production must at all times be equal to consumption. That is to say, the market must balance. To make it easier to balance the market, five price areas have been created for electricity. The areas No1, No2, and No5 are in southern Norway, while No3 and No4 are in northern Norway [25]. In practice, we have seen that there is little price difference between the three price areas in southern Norway (No1, No2, and No5) and the two price areas in northern Norway (No3 and No4). Therefore, in this article we will combine the price areas in the south and in the north and refer to these as price in southern Norway ($p_{sn}$) and price in northern Norway ($p_{nn}$), respectively. If $p_i$ is the electricity price in area $i$ we have:

$$p_{sn} = (p_{no1} + p_{no2} + p_{no5})/3 \qquad p_{nn} = (p_{no3} + p_{no4})/2$$

Figure 10 presents the electricity prices in southern Norway ($p_{sn}$) measured in euros since the year 2000. The data reveal that from 2000 to 2020, the prices remained relatively stable and low. During the period of 2013 to 2020, the average price was around NOK 286 per MWh. However, a dramatic increase occurred in the autumn of 2020, with prices peaking at NOK 7820 (approximately EUR 780) on 30 August 2022. Such significant fluctuations in the price of a necessity like electricity are quite rare.

While numerous studies have been conducted on the price elasticity of electricity [26,27], there appears to be a lack of research into scenarios where price changes are as substantial and abrupt as those experienced in southern Norway. Therefore, we aim to explore this phenomenon in greater detail.

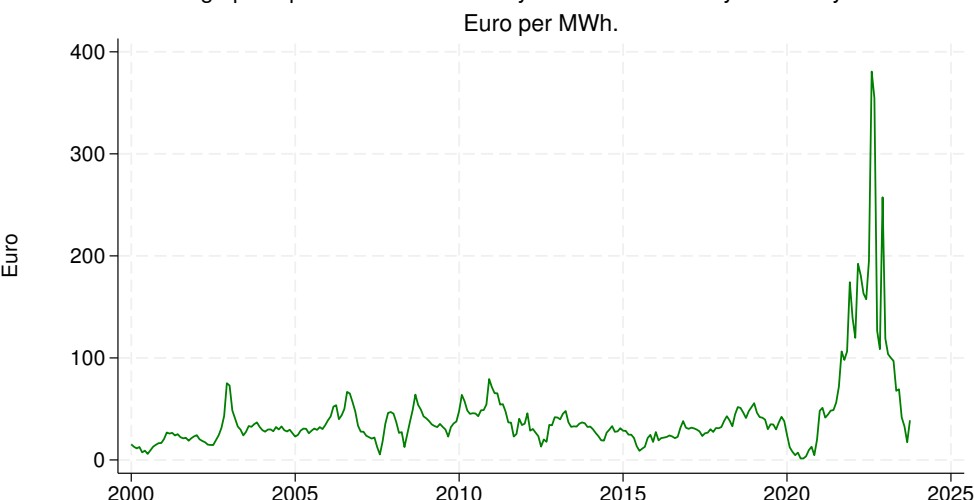

**Figure 10.** The price of electricity in euros per MWh in southern Norway in the period 2000–2023.

Price Differences between Northern Norway and Southern Norway

Norway is an elongated country with limited transmission capacity of electricity between north and south. Although the transmission capacity between northern and southern Norway is small, the price of electricity in the two regions was more or less the same until the year 2020. This can be seen from Figure 11.

In recent years, there have been significant changes in Europe's electricity market. This is due to the expansion of wind power. Germany, the United Kingdom, and Spain are at the forefront of this development. In Germany, 551 wind turbines were installed in 2022, with a total installed capacity of 2.4 GW.

According to the industry association Wind Europe, several thousand new turbines are planned to be installed in Europe between 2022 and 2027. Additionally, offshore wind projects with an annual capacity of 1.3 GW are also anticipated during this period [28].

Uncontrollable wind power increases the need for stable, controllable balancing power such as hydropower. This increased the need for both Germany and England to import power from Norway. The cable to Germany came into full operation in April 2021, while the cable to England came into operation in October 2021. Before the last two cables were built in 2021, there was little difference in the price of electricity between southern Norway and northern Norway (see Figure 11). As can be seen from Table 1 and Figure 12, this has changed. In Norway, a new term was used to describe the situation. It was price contagion. northern Norway was shielded from price contagion due to small cable capacity between northern Norway and southern Norway.

**Table 1.** The average price of electricity in NOK per MWt in northern Norway and southern Norway in the periods 2013–2020 and 2020–2023. Last day of observation: 15 November 2023.

|  | **2013–2020** | **2013–2020** | **2020–2023** | **2020–2023** |
|---|---|---|---|---|
|  | Price | St.dev | Price | St.dev |
| Southern Norway | 262.7 | 116.1 | 1196.7 | 928.1 |
| Northern Norway | 269.3 | 112.4 | 356.1 | 353.6 |
| Difference | −6.6 | 33.9 | 841.0 | 881.2 |

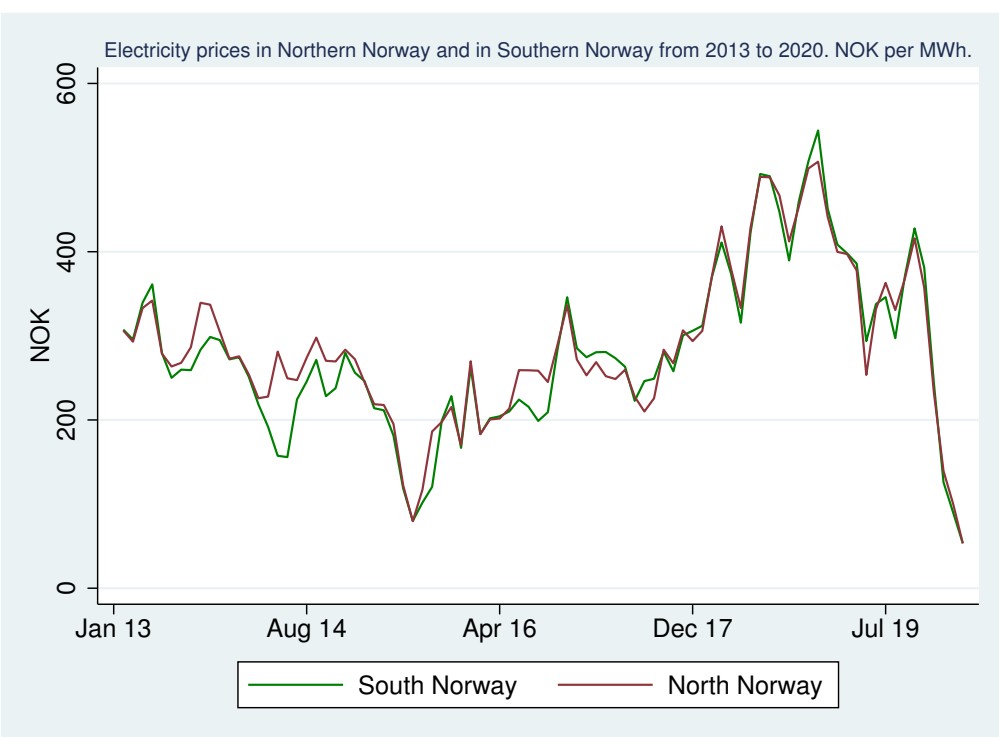

**Figure 11.** The price of electricity from northern Norway and southern Norway from 2013 to 2020. During this period, prices were more or less the same in the two regions.

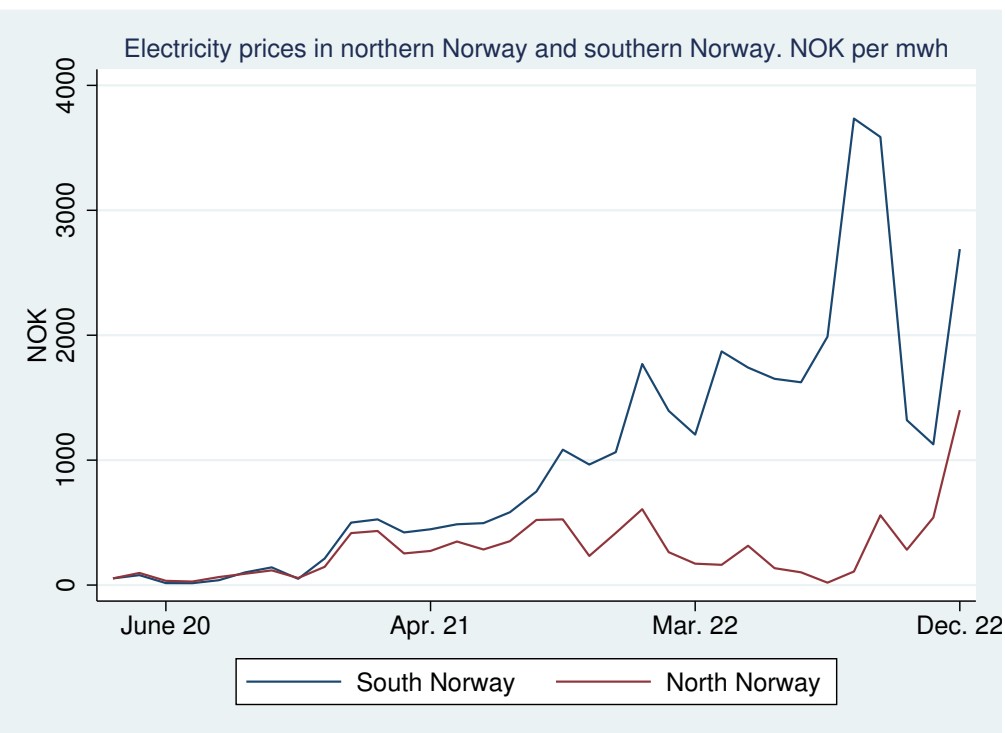

**Figure 12.** The price of electricity in northern Norway and southern Norway after the year 2020.

## 2. Research Question and Method

Our research question addresses a significant economic event: in the autumn of 2020, southern Norway experienced a substantial surge in electricity prices. While the average price from 2013 to 2019 was NOK 263 per MWh, it escalated to NOK 1192 per MWh during

2020–2023. We aim to investigate the impact of this price hike on electricity consumption patterns.

To answer our questions, we will formulate a log-linear model where the consumption of electricity in southern Norway *C* is dependent on three explanatory variables: the price of electricity (P), the temperature (*T*), and wind speed (*V*). Since it is the price elasticity we want to investigate, the price must necessarily be included as an explanatory variable in the model. Here, we use the Nord Pool spot price even though the authorities introduced a power support scheme in 2022 for consumption in homes and for businesses. We will return to this support scheme below in Section 3.3. Wind speed and temperature are the other two explanatory variables requires an explanation:

1. Energy traders who buy and sell electricity contracts in the day-ahead market use temperature and wind strength as two key variables when making their decisions [29].
2. Temperature is included as an explanatory factor because Norway is a country with long and cold winters, and many people use electricity to heat their homes. According to Statistics Norway, 75 percent of the energy used for heating in Norway is based on electricity. In the EU, the corresponding proportion is around 5 per cent [30]. Wind speed is included as an explanatory factor because strong winds combined with low temperatures contribute to cooling down the houses.

Since electricity consumption fluctuates greatly throughout the year, should not the model reflect this by using dummy variables? We believe that with our explanatory variables, it would be incorrect to use dummy variables to capture seasonal variations. Consumption in January is high not because the name of the month is January, but because it is cold.

We assume that there is the following relationship between consumption and the explanatory variables:

$$C = f(P, T, V) = A \cdot P^{b1} \cdot T^{b2} \cdot V^{b3} \tag{1}$$

where *P* is price of electricity in southern Norway, *T* is temperature measured in Fahrenheit (to be able to use logarithms), *V* is wind speed and *A*, *b1*, *b2*, and *b3* are positive constants. Equation (1) is a special case of a Cobb–Douglas function. When conducting a regression analysis, choosing a model can be problematic, but from Figure 13, there is reason to believe that the relationship indicated in (1) fits well in our case.

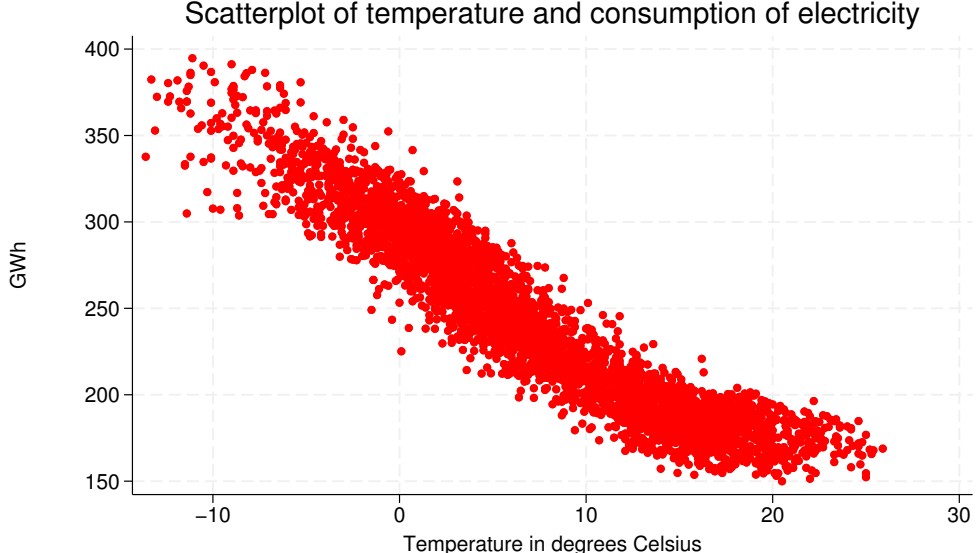

**Figure 13.** Scatter plot of consumption and temperature. The graph indicates that there is a linear relationship between consumption and temperature.

By finding the partial derivatives of (1) and substituting this into the definition of elasticity, we find that the exponents *b1*, *b2*, and *b3* are the partial elasticities.

Based on Equation (1), we can now establish the following linear regression model:

$$\ln C_i = Z + b_1 \ln P_i + b_2 \ln T_i + b_3 \ln V_i + \mu_i \tag{2}$$

where $Z = \ln A$, $\mu$ is the error term.

Equation (2) is linear in the parameters. We used Stata to estimate the regression coefficients $b1$, $b2$, and $b3$. A one percentage change in one of them gives the percentage change in the consumption $(C)$, ceteris paribus. This means that $b1$ is the estimated price elasticity, $b2$ is the estimated wind speed elasticity, and $b3$ is the estimated temperature elasticity. We used data from Nord Pool [31] and The Norwegian Meteorological Institute [32].

We have created a model where we estimate three elasticities. The price elasticity, that is, the percentage change in consumption when the price changes by one percent, is quite common to estimate, but the last two elasticities, wind speed elasticity and temperature elasticity, are not common to estimate; therefore, we would like to make some remarks. Elasticities are practical to use as they make it possible to compare between markets and products. What happens to consumption when the wind speed increases or decreases by one percent, and what happens to consumption when the temperature rises or falls by one percent, are questions of the type that are important to be able to answer if one wants to understand how the electricity market in the Nordic countries reacts to changes in weather conditions. For that reason, we have used elasticities.

*Modeling Consumption as a Time Series*

Above, we have considered our data as cross-sectional data, but in cases where consumption fluctuates with the seasons, such as in Norway, consumption can also be modeled as a sine curve with a trend coefficient. See Figure 6 in Section 1.4. Here (Figure 14) below, is a proposal that can be used to give an approximate forecast of consumption in these cases:

$$y(t) = A \cdot \sin(B \cdot t + C) + D + k \cdot t \tag{3}$$

where $A$ is the amplitude (the difference between peak and average value), $B$ is the frequency of the curve, $C$ is the phase (the displacement of the curve along the $x$-axis), $D$ is vertical displacement (average value), $t$ is the time (in our case, days), and $k$ is a trend coefficient. In our case: A=100,000, D=250,000, $B = 2\pi/365$, $k = 0$, and $C = (-2\pi/365 - 5)$.

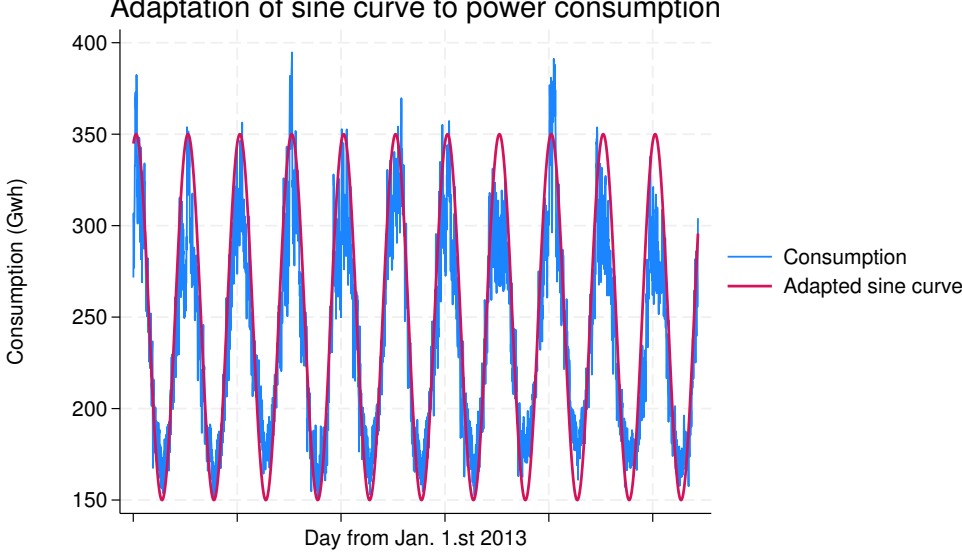

**Figure 14.** The consumption of electricity modeled as a sine curve with a trendfactor of zero.

In our data, there is no upward or downward trend. This means that the trend factor is equal to zero. But a transition from petrol or diesel-powered cars and construction machinery to battery-powered ones will make it necessary so set k > 0 in the future.

## 3. Results and Discussion

The result of our regression is given in Figure 15:

| Source | SS | df | MS | | |
|---|---|---|---|---|---|
| | | | | Number of obs = | 3,967 |
| | | | | F(3, 3963) = | 7905.42 |
| Model | 169.913396 | 3 | 56.6377988 | Prob > F = | 0.0000 |
| Residual | 28.3926246 | 3,963 | .007164427 | R-squared = | 0.8568 |
| | | | | Adj R-squared = | 0.8567 |
| Total | 198.306021 | 3,966 | .050001518 | Root MSE = | .08464 |

| lnC | Coefficient | Std. err. | t | P>|t| | [95% conf. interval] | |
|---|---|---|---|---|---|---|
| lnP | -.0087249 | .0014413 | -6.05 | 0.000 | -.0115506 | -.0058992 |
| lnV | .0221569 | .0032601 | 6.80 | 0.000 | .0157653 | .0285485 |
| lnT | -.6029438 | .0039871 | -151.23 | 0.000 | -.6107607 | -.5951269 |
| _cons | 14.66347 | .0183296 | 799.99 | 0.000 | 14.62754 | 14.69941 |

**Figure 15.** The solution when we assume that the model given in Equation (2) fits in our case.

The solution given in Figure 15 may seem convincing at first glance. All three explanatory variables are significant; the signs of the regression coefficients are in line with economic theory and $R^2 = 0.86$. When the price rises, consumption falls, but the price elasticity ($-0.087$) percent is very small and indicates inelastic price elasticity. As the wind speed increases, the consumption also increases. The wind speed elasticity is relatively small, at only 0.022; however, its impact is statistically significant with a t-value of 6.8.

The estimated temperature elasticity is $-0.603$. If the temperature increases by one percent, the consumption decreases by 0.6 percent. Variations in temperature therefore have a great impact on variations in consumption.

Our model assumes constant elasticities across the entire dataset, which is a condition we cannot verify initially. To investigate potential variations in elasticities, we divided the dataset into four segments. Our model initially considers a three-dimensional space defined by three explanatory variables. For simplicity, we reduced this to a two-dimensional plane, focusing on key variables by setting specific limits for price and temperature. We categorized the data points based on these limits: electricity prices above 1500 NOK per MWh are labeled 'high price', and temperatures above 0 °C are considered 'warm'. This segmentation allows us to examine if and how the elasticities differ across these defined categories:

1. High price and warm: P > 1500 and T > 0. Result: See Figure 16, northeast quadrant;
2. High price and cold: P > 1500 and T < 0. Result: See Figure 16, southeast quadrant;
3. Low price and warm: P < 1500 and T > 0. Result: See Figure 16, northwest quadrant;
4. Low price and cold: P < 1500 and T < 0. Result: See Figure 16, southwest quadrant.

By dividing the dataset in this manner and utilizing the model described in Equation (2), we obtain the results that are depicted in Figure 16:

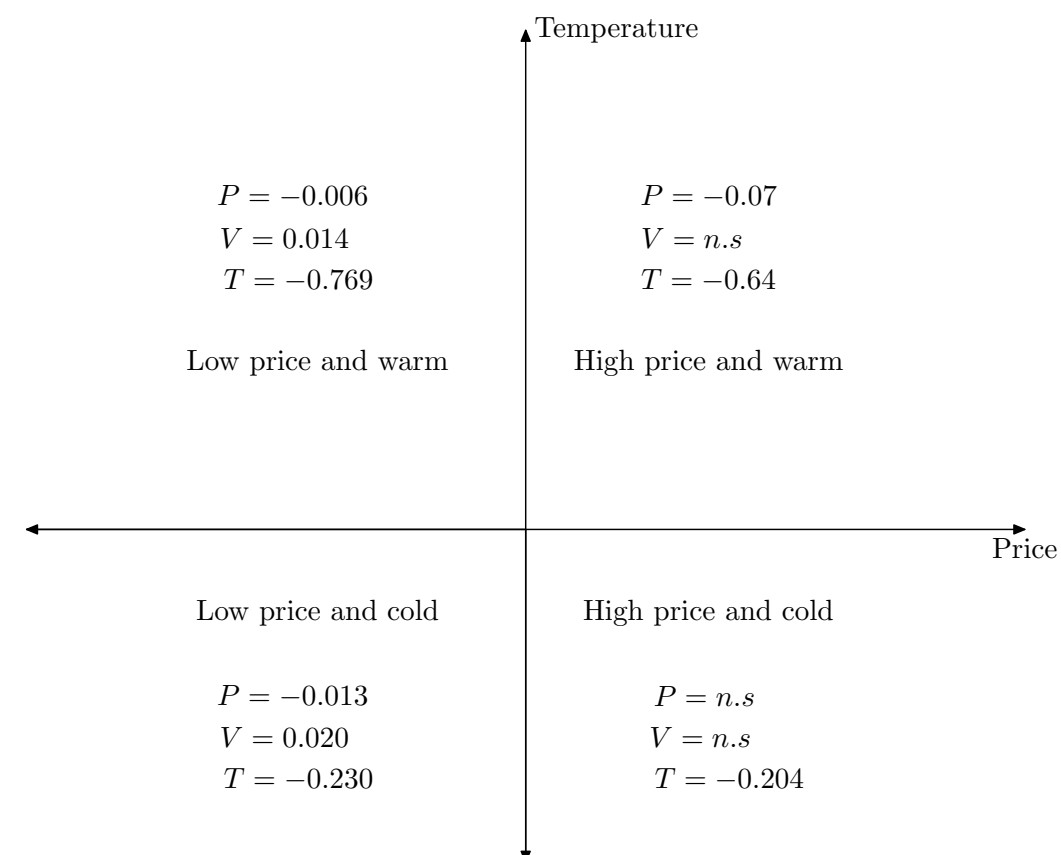

**Figure 16.** The estimated regression coefficients (the elasticities) in the four groups mentioned above. n.s means not statistically significant.

From Figure 16, we see that the price of electricity and wind speed have little significance on consumption. What matters is the temperature. However, as shown in Table 2 below, the temperature elasticity is not constant. For example, changes in electricity consumption are greater with temperature changes when the temperature is $+10\,°C$ than when the temperature is $-10\,°C$.

**Table 2.** Temperature elasticity in different temperature categories.

| Temperatures (°C) | Elasticity | t-Value | $R^2$ | Observations |
|---|---|---|---|---|
| Temperature > 5 | −0.73 | −79.4 | 0.74 | 2315 |
| −2 < Temperature < 5 | −0.55 | −28.8 | 0.42 | 1184 |
| Temperature < −2 | −0.19 | −18.3 | 0.44 | 437 |

Why is temperature elasticity not constant? We are not aware of any research in this field, but would like to put forward two possible explanations:

1.  Heating with wood: Apart from electricity, wood is the most important source of energy for heating in households. Firewood accounted for 16 percent of supplied energy in 2012 [33]. Heating the home with wood involves costs. If we disregard the purchase costs, wood burning implies a lot of practical work: the wood must usually be collected from a warehouse outside the house, using the wood will lead to a little more dust in the living room, the top must be taken care of so that the air supply is just right, and after a few days of burning wood, the ashes must be removed from the stove. This additional work means that wood burning is something that people in Norway only do when it is absolutely necessary to avoid freezing (there are of course exceptions). For most people, the most convenient thing is to use electricity. But when

it gets very cold, for example colder than $-15\,°C$, the capacity of the electricity in many homes is not good enough. If it is very cold, a little more electricity is used, but even more wood is used.

When wood burns, it incurs external costs since it deteriorates air quality. But in 1998, the requirement was introduced that all new wood stoves sold and installed in Norway must be clean-burning for the sake of the climate. Wood stoves produced before 1998 emit significantly more particulate matter than modern stoves. Switching from a non-clean-burning to a clean-burning wood stove will reduce the amount of particulate matter by a notable 90 percent.

2. Use of heat pumps: In 2021, 39 per cent of households in Norway had a heat pump [34]. It is well understood that, in general, air source heat pump efficiency decreases with decreasing outdoor temperature [35]. How much efficiency to heat pumps increases with higher temperature depends on the type of heat pump. An example from the literature states that the COP (coefficient of performance) at $-12\,°C$ was equal to 1.33, while it was 2.9 at $+15\,°C$ [36].

### 3.1. Regression with Standardized Variables

Which explanatory variable is most significant in explaining changes in consumption? A common method to answer this question is to standardize the explanatory variables [37]. This means that we transform each explanatory variable as follows:

$$z = \frac{x - \bar{x}}{\sigma_x} \tag{4}$$

where $z$ is the standardized value of $x$ and $\bar{x}$ is the mean of $x$ and $\sigma_x$ is the standard deviation of $x$. The standardized variable $z$ has a mean of 0 and a standard deviation of 1. By doing this for all three explanatory variables, it becomes easier to determine which variable is most significant in explaining changes in the dependent variable. We denote the standardized explanatory variables as follows: standardized price of electricity in southern Norway $p_s$, standardized wind speed $v_s$, and standardized temperature $t_s$. Our model then becomes:

$$C_i = \beta_0 + \beta_1 p_s^i + \beta_2 v_s^i + \beta_3 t_s^i + \epsilon_i \tag{5}$$

When we perform the regression with standardized explanatory variables, we obtain the result as shown in Figure 17:

```
. reg C p_s v_s t_s
```

| Source | SS | df | MS | | | |
|---|---|---|---|---|---|---|
| | | | | Number of obs | = | 3,970 |
| | | | | F(3, 3966) | = | 11422.51 |
| Model | 1.0330e+13 | 3 | 3.4433e+12 | Prob > F | = | 0.0000 |
| Residual | 1.1955e+12 | 3,966 | 301447379 | R-squared | = | 0.8963 |
| | | | | Adj R-squared | = | 0.8962 |
| Total | 1.1525e+13 | 3,969 | 2.9039e+09 | Root MSE | = | 17362 |

| C | Coefficient | Std. err. | t | P>\|t\| | [95% conf. interval] | |
|---|---|---|---|---|---|---|
| p_s | -2358.813 | 276.6724 | -8.53 | 0.000 | -2901.246 | -1816.379 |
| v_s | 605.9753 | 276.8603 | 2.19 | 0.029 | 63.17331 | 1148.777 |
| t_s | -51104.34 | 276.3497 | -184.93 | 0.000 | -51646.14 | -50562.54 |
| _cons | 239263.8 | 275.5564 | 868.29 | 0.000 | 238723.5 | 239804 |

**Figure 17.** The result of regression with standardized variables as explanatory variables.

The coefficient 'cons' in Figure 17 is an estimate of $\beta_0$ in Equation (5). We obtained $\hat{\beta}_0 = 239,263.8$, which is close to the average value of daily consumption ($\bar{C} = 239,258.8$) during the observation period 2013–2023. The coefficient 'cons' represents the expected

value when all explanatory variables are equal to zero. Since the variables are standardized, and thus have a mean value of zero, this would correspond to the expected value of the dependent variable when all explanatory variables are equal to their mean value.

All the estimated coefficients $\hat{\beta}_1$, $\hat{\beta}_2$, and $\hat{\beta}_3$ are significant (see the t-value and confidence interval). The estimates of the coefficients for the three explanatory variables are $\hat{\beta}_1 = -2359$, $\hat{\beta}_2 = 606$, and $\hat{\beta}_3 = -51{,}104$. Since the explanatory variables are standardized, the coefficients represent the change in the dependent variable when the explanatory variable increases by one standard deviation.

### 3.2. Validity of the Results

Whether the estimated results are valid or not depends on several factors. Therefore, we conducted three tests:

1.  Multicollinearity Test. We started by checking for multicollinearity among the explanatory variables, as high correlation between the variables can distort the estimates. In this test, we calculated the extent to which one independent variable can be explained by the other independent variables. This is achieved by calculating the VIF (Variance Inflation Factor). The VIF for the independent explanatory variable a is defined as follows:

$$VIF_a = \frac{1}{(1 - R_a^2)} \tag{6}$$

By conducting an auxiliary regression where one of the explanatory variables is chosen as the dependent variable while the other right-hand side variables serve as explanatory variables, one can calculate $R^2$. An $R^2$ value close to zero indicates that there is little correlation between the explanatory variables. This results in a VIF (Variance Inflation Factor) close to 1. In our case, we obtained the following VIF:

| Variable | VIF | 1/VIF |
|---|---|---|
| v_*s* | 1.01 | 0.990611 |
| p_*s* | 1.01 | 0.992005 |
| t_*s* | 1.01 | 0.994517 |
| Mean VIF | 1.01 | |

From the table above, we see that the VIF factor is close to one. Therefore, there is no multicollinearity in our case.

2.  Heteroskedasticity Test: Heteroskedasticity is a problem because it can lead to the standard errors of the regression coefficients being unreliable. Consequently, we cannot rely on *t*-tests regarding the significance of the coefficients, and the confidence intervals also become incorrect. Heteroskedasticity can be due to errors in model specification, such as omitted variables or incorrect functional form.

    In our case, we tested for heteroskedasticity using the Breusch–Pagan test [37]. The null hypothesis is: There is no heteroskedasticity. We choose a significance level of 0.05. The test must then yield a *p*-value that is lower than 0.05 for us to reject the null hypothesis. In our case, we obtained a *p*-value of 0.24. Since the *p*-value is greater than the chosen significance level, we cannot reject the null hypothesis. Therefore, we assume that there is homoskedasticity.

3.  Normality of the residuals test: Our case does not satisfy the requirement for the residuals to be normally distributed. In our situation, we have observations for all variables—the dependent and the three independent ones—over 3970 days. Based on the Central Limit Theorem, we therefore choose to assert that the results we have arrived at are valid [38]. This is also confirmed by Figure 18.

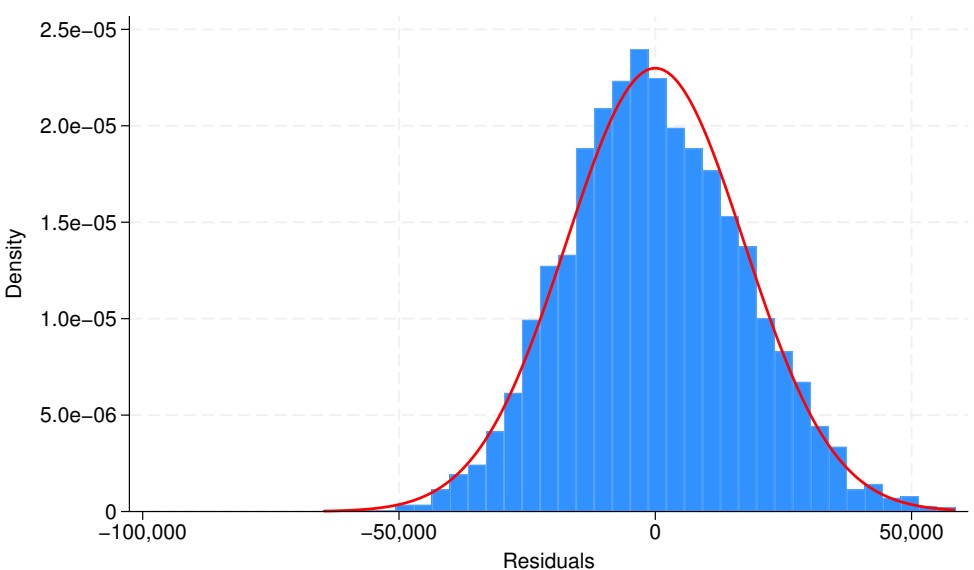

**Figure 18.** Residuals and a normal distribution plotted in the same diagram.

### 3.3. Some Remarks on Support for Businesses and Consumers

The authorities introduced electricity support for consumers' residential consumption in December 2021. Residential homes use 24.8 percent of the electricity produced in Norway (based on average consumption and production in the years 2021, 2022, and 2023) [39]. The price that consumers pay is the Nord Pool spot price plus taxes and value-added tax (VAT) and grid rental minus government support. The support amounts to 90 percent of the amount by which the price exceeds NOK 730 per MWh. Electricity support is given for consumption lower than 5000 kWh per month. Electricity support is not given for the consumption of electricity in holiday homes. Consumers must pay two types of taxes. There is the value-added tax of 25 percent. In addition, there is an extra tax of NOK 10 per MWh. In addition to this, consumers must pay grid rental. The grid rental consists of a fixed and a variable part. The average grid rental in 2024 is NOK 535.5 per MWh [40].

A study of what proportion of income is spent on covering electricity costs for people is a larger project that is outside our topic. However, by using information from Statistics Norway, we can arrive at some average figures. An average household in Norway consists of 2.11 persons. Each household uses 18,100 kWh of electricity per year. The average wage income in Norway was NOK 668,400 per year in 2023. Assuming a Nord Pool price of electricity at NOK 1000 per MWh, a household with one income earner will spend 5.2 percent of their income on electricity (including taxes and grid rent) if they do not receive support from the authorities. If they receive support, the proportion of income spent on electricity would decrease to 4.4 percent

In the winter of 2022, the authorities also established a support scheme for businesses. Businesses were then able to apply for grants that provide 25 percent support for electricity prices over NOK 700 per MWh. In 2022, support was given to 3200 businesses. There were then 100,460 businesses in Norway with more than five employees. Assuming that the businesses eligible to apply were those with more than five employees, this means that 97 percent of businesses did not receive support. Figure 19 shows the electricity costs after taxes and grid rent.

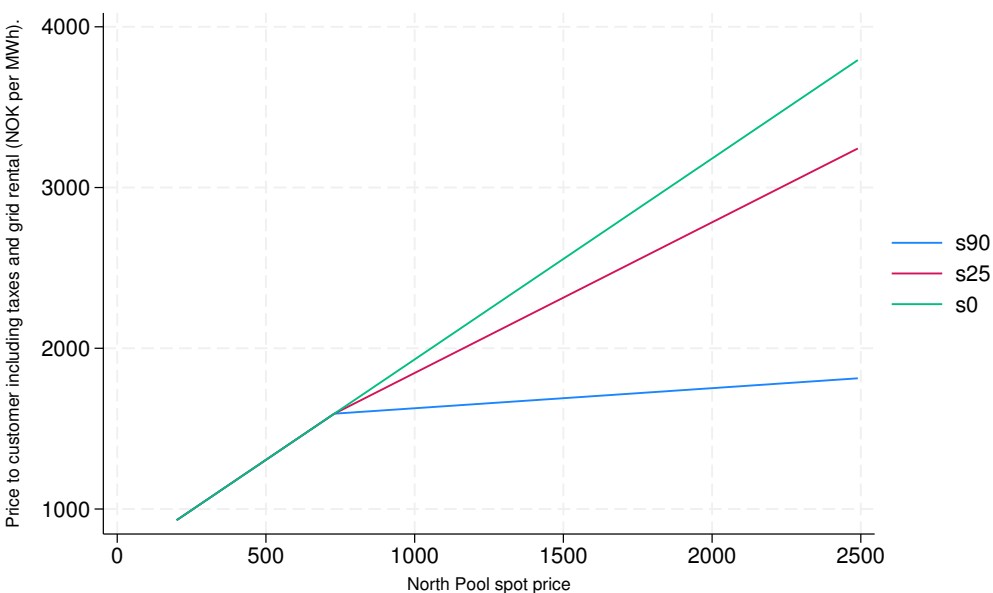

**Figure 19.** The lines s0, s25, and s90 represent the costs for customers who did not receive support, the costs for customers who received 25 percent support, and customers who received 90 percent support, respectively. If the Nord Pool price was lower than NOK 700 per MWh, no businesses received support. If the Nord Pool price was lower than NOK 730 per MWh, no one received support for consumption in homes. Most businesses (97 percent) had costs following s0.

## 4. Conclusions

The Norwegian electricity market distinguishes itself from other countries' energy markets in several notable ways:

- Virtually all electricity comes from renewable sources (as of 2023, mainly hydropower and wind power);
- In total, 75 percent of the energy used for heating comes from electricity;
- In years with a normal amount of rainfall, Norway is a net exporter of electricity to Sweden, Denmark, Finland, the Netherlands, England, and Germany;
- Excluding electricity, wood is the most important energy source for heating homes;
- In total, 39 percent of residential houses in Norway use heat pumps.

Therefore, there is a risk that conclusions about the Norwegian electricity market, based on research conducted in other countries, may be incorrect or inaccurate.

Since the fall of 2020, electricity prices in Norway have more than quintupled compared to previous years. This price increase provided an opportunity to calculate price elasticity, temperature elasticity, and wind speed elasticity. The results were as follows:

1. Price elasticity is not significant when the temperature is below 0 °C and the price is higher than NOK 1500 per MWh. Outside this range, the price elasticity is very low, at only −0.01. Within the significant range, a higher price leads to slightly lower consumption.
2. Wind speed elasticity is not significant when the price is higher than NOK 1500 per MWh. When the price is lower, wind speed elasticity is very low, between 0.01 and 0.02. In the significant range, higher wind speed slightly increases consumption.
3. Temperature elasticity is significant at all temperatures and all price levels. This means that consumption increases as the temperature drops and vice versa. Temperature elasticity is not constant; it is approximately −0.7 when the temperature is above 0 °C and −0.2 when below 0 °C. Therefore, the change in consumption per degree change in temperature is significantly higher when temperatures are above freezing than when they are below. The reason for higher temperature elasticity at higher temperatures compared to lower temperatures requires further research. A possible hypothesis is the effect of wood-burning and the use of heat pumps.

In cold climates like in Norway, heating of homes is a necessity to maintain a livable environment. In the ERG theory of human motivation [41–43], this falls under the Existence category, which is about basic physical needs. Making changes to electricity consumption takes time and usually requires investments. Therefore, our study is only valid in the short term.

**Author Contributions:** All authors have contributed to writing—review and editing; methodology, J.I. All authors have read and agreed to the published version of the manuscript.

**Funding:** This research was funded by Western Norway Research Institute.

**Institutional Review Board Statement:** Not applicable.

**Informed Consent Statement:** Not applicable.

**Data Availability Statement:** Weather data can be downloaded for free from Norsk klimaservicesenter https://seklima.met.no/observations/. Data on electricity prices and consumption can be purchased from Nord Pool https://www.nordpoolgroup.com/.

**Conflicts of Interest:** The authors declare no conflicts of interest.

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
