# Peer review of "The Short-Term Price Elasticity, Temperature Elasticity, and Wind Speed Elasticity of Electricity: A Case Study from Norway"

_sustainability, doi:10.3390/su16083321_

Round 1
Reviewer 1 Report
Comments and Suggestions for Authors
The paper describes the characteristics of the Norwegian electricity market and the effects of temperature and wind speed on the price of electricity consumption.
The paper is written in the form of a report. Very convenient if you want to know the Norwegian electricity market. There is no research proposal, only analysis of the factors that influence the price of electricity. Elasticity is used to determine the influence of temperature on the price of electricity.
Some suggestions:
A text describing the content of the article is missing. A section that describes the methodology (methods and materials) is missing.
Improve the explanation of the concept of estimated price elasticity, estimated wind speed elasticity, and estimated temperature elasticity.
Improve the explanation of figure 16.
Author Response
Thank you for taking the time to give us good advice. We have changed the article and taken your advice into account.

Reviewer 2 Report
Comments and Suggestions for Authors
General Comments
The authors conducted an interesting case study on the increase in electricity prices in the Norwegian market as a result of newly constructed cross-border connections during the price crisis in European markets caused by the cessation of gas imports from Russia. The reason for the price increase in Norway seems to be the opening of the market to exports to countries where high marginal prices are observed. This made it possible to shape prices at the level of the most expensive sources in the EU countries to which this energy is supplied. Therefore, cross-border connections probably led to price reductions in countries with high price levels and to price increases in Norway as a result of the use of "market coupling" mechanisms with power lines having limited capacity.
The authors presented a mathematical model of the influence of temperature, wind, and price on the level of consumption. The result is low price elasticity of customer demand. The response to changes in energy prices is rather long-term because, as the authors note in the last sentence of their article, the change in the source of investment expenditure. However, the recent reduction in prices - as shown in Fig. 10 - may slow down adjustment activities.
In this aspect, it would be important to show how burdensome is the energy price level for the inhabitants of Norway, e.g. in the form of the average expenditure for energy of the inhabitants in relation to their income - the level of 10% is considered to be the limit of "energy poverty". However, the increase in the use of wood indicates a certain nuisance due to the significantly increased level of energy prices.
The authors do not analyze the level of retail market prices, only wholesale prices. Therefore, it requires an explanation of how wholesale prices translate into retail prices in Norway. Do retail prices dynamically keep up with wholesale prices and whether any mechanism has been created to protect residents and business customers against increases in energy prices. The EU even introduced recommendations to tax energy suppliers to redistribute their excess profits to consumers. The high profits of energy-generating companies in Norway, taking into account their low production costs, would justify the introduction of such mechanisms.
An increase in prices, previously low due to the use of renewable energy to produce electricity, may result in the desire to use cheaper energy carriers, which, however, cause pollution, which should burden their users with the costs of reducing air quality in the long term.
Detailed comments
Line 56
'The second advantage is the minimal wear and tear on the production equipment, resulting in low production costs'.
Low prices are also determined by the lack of need to purchase fuel.
Figure 9. Standardized share of exchange and production of electricity
It would be advisable to present what relationships express the "Standardized share of exchange and production of electricity"
Editorial errors noted
Line 180
1.6.1. Price differences between northern Norway and southerb Norway
Line 239
but but in cases where
Author Response

(The authors gave the same response as above.)

Round 2
Reviewer 2 Report
Comments and Suggestions for Authors
Thank you for the additions to the text and additional explanations. They allow a better understanding of the reactions of electricity recipients to changes in the market conditions.